# Spatio-temporal characterization of earthquake sequence parameters and forecasting of strong aftershocks in Xinjiang based on the ETAS model

**Ke Li[1], Maofa Wang[2], Huiguo Zhang[1], Xijian Hu●[1] ***

**1** College of Mathematics and System Science, Xinjiang University, Urumqi, Xinjian, China, **2** Guangxi Key Laboratory of Trusted Software, Guilin University of Electronic Technology, Guilin, China

* xijianhu@xju.edu.cn

**Data Availability Statement:** All seismic data of Xinjiang region are available from the https://data.earthquake.cn/datashare/report.shtml?PAGEID=earthquake_zhengshi.

## Abstract

In this paper, the Integrated Nested Laplace Algorithm (INLA) is applied to the Epidemic Type Aftershock Sequence (ETAS) model, and the parameters of the ETAS model are obtained for the earthquake sequences active in different regions of Xinjiang. By analyzing the characteristics of the model parameters over time, the changes in each earthquake sequence are studied in more detail. The estimated values of the ETAS model parameters are used as inputs to forecast strong aftershocks in the next period. We find that there are significant differences in the aftershock triggering capacity and aftershock attenuation capacity of earthquake sequences in different seismic regions of Xinjiang. With different cut-off dates set, we observe the characteristics of the earthquake sequence parameters changing with time after the mainshock occurs, and the model parameters of the Ms7.3 earthquake sequence in Hotan region change significantly with time within 15 days after the earthquake. Compared with the MCMC algorithm, the ETAS model fitted with the INLA algorithm can forecast the number of earthquakes in the early period after the occurrence of strong aftershocks more effectively and can forecast the sudden occurrence time of earthquakes more accurately.

## Introduction

Earthquakes, as the most destructive natural hazard, can have devastating consequences. Point process modeling can be used to better understand potential interactions between earthquakes and spatial features, temporal features, and the effects of other covariate effects. Each earthquake event can be viewed as a point related to space (latitude and longitude coordinates), time of occurrence, and magnitude. Anwar (2009) [1] and Ye (2015) [2] combined a multi-type Strauss process and geospatial covariates combined to describe earthquake sequences. Siino et al. (2017) [3] used the Gibb mixing model to describe seismicity in Greece, showing the interaction of seismicity at different spatial scales (multiscale structure), characterizing the spatial heterogeneity of the process. All the above modeling approaches are used to describe

**Funding:** This study was supported by the National Natural Science Foundation of China (11961065, 42164002), The Ministry of Education of Humanities and Social Science project (19YJA910007), and the Natural Science Foundation of Xinjiang (2023D01C01).

**Competing interests:** The authors have declared that no competing interests exist.

the evolution of earthquake occurrence in both spatial and temporal dimensions. As for the Cox process model [4], which can often be used to characterize environmentally driven processes (Møller et 2003) [5], the log Gaussian Cox process (LGCP) is a special case of the Cox process used to characterize that the rate of occurrence of certain earthquakes will be determined by a potential random field. Bayliss (2020,2022) [6, 7] used synthetic and real data from California earthquakes to characterize the intensity of spatial variability in earthquake locations through a log-Gauss-Cox process model and evaluates the performance of models of varying complexity consisting of different components to determine which elements are most useful for describing the distribution of earthquake locations.

After the occurrence of a moderate-to-strong earthquake, the characterization of the aftershock sequence after the main earthquake plays a key role in determining the type of earthquake sequence and forecasting the likelihood of strong aftershocks. The rate of decay in event rate during aftershock sequences and the ability to stimulate aftershocks can be characterized by the statistical parameters of the sequences. Therefore, obtaining accurate and reliable earthquake sequence parameters is an important reference value for quickly determining the type of earthquake sequence, studying the characteristics of the seismic source region, and forecasting subsequent earthquakes. In 154 regions of China, Haikun Jiang (2007) [8] examined the features of the ETAS model parameters in the early period of the aftershock of moderate-to-strong earthquake sequences. The study focused on comparing the relative sizes of the two crucial parameters, $p$ and $\alpha$, which have distinct physical meanings and are crucial to the model's operation. The study also discussed the practical issues of the aftershock sequence's decay in event rate and the probability that one event will cause an aftershock. Similarly, estimating the likelihood of earthquakes is the core of earthquake forecasting research and is important for earthquake risk management, especially during sustained earthquake sequences. The epidemic-type aftershock sequence model (ETAS) is widely used to model earthquake sequences [9–11] and also provides an effective tool for forecasting the spatiotemporal evolution of short-term aftershock clusters [12]. The ETAS model is the current benchmark for operational earthquake forecasting [13], showing that the ETAS model was then the best model for forecasting short-term seismic activity. In recent years of research, Yaghmaei-Sabegh S et al [14] modeled the hierarchical structure of aftershock sequences using the epidemic-type aftershock sequence (ETAS) model and compared it with the RJ model. In this paper, they also used a smoothing-based approach to incorporate the statistical modeling of aftershocks into the framework of Probabilistic Seismic Hazard Analysis (PSHA) [15], which is used to determine the probability that a given level of ground motion will be exceeded over a given time horizon and is a fundamental tool in many seismic design codes. The results show that the ETAS model with higher PGA values better simulates the behavioral patterns of aftershock sequences with branching structure than the RJ model that only considers primary aftershocks. The ETAS model, also known as the Hawkes process in statistics, is a self-excited point process model that describes a sequence of earthquakes that occur in space-time and space-time. Each occurrence of the event triggers the probability of a sub-event occurring, and at the same time, this triggering function decays in space-time [16].

In a Bayesian framework, Bachl et al. proposed a method for approximating the parameter posterior via the R-Inlabru package based on the Integrated Nested Laplace Approximation (INLA) method (Rue 2018) [17]. The INLA method is an efficient alternative to the MCMC method used to compute Latent Gaussian Models (LGMs) (Robert 2014) [18]. When it comes to fitting model parameters, the primary distinction between these two approaches is that the INLA method is quicker because it employs a deterministic approximation, while the MCMC method is based on simulation. The INLA algorithm's drawback is that it is unable to handle completely non-linear problems. Because of its higher computational cost, the MCMC method

finds it difficult to achieve modeling in the presence of strongly correlated parameters, a situation that will arise in many applications and for which the INLA method is intended to assist. The INLA algorithm has demonstrated its benefits in several research regions, e.g., disease mapping [19, 20], genetics [21], public health [22], ecology [23], and forecasting seismicity [24].

The Xinjiang seismic zone has five major seismic zones and is one of the most seismically active regions in China. At the same time, Xinjiang is a remote and poor region in China, the urban and rural economy is backward, houses, structures, and other infrastructure are poor, resulting in several casualties and large economic losses during strong earthquakes. Since 1996, more than 10 consecutive earthquakes of magnitude 6 or above occurred in Gashi-Atushi and other places, which aroused the attention of the relevant experts in China. Gao Yalan (2023) [25] constructed an ARIMA forecasting model to forecast the trend of crustal change based on the deformation data at the whole point of time of the Jinghe seismic station monitoring in the past ten years in Xinjiang region, to explore the kinematic law of crustal deformation in Jinghe. The Markov chain method was employed by Zhang Linlin (2012) [26] to forecast the magnitude 1 to 4 earthquakes that occurred in various Xinjiang regions between 2000 and 2004. The method is practical and effective for use in earthquake forecasting, and the forecasting parameters were regularly verified and adjusted using the region's earthquake data.

To accurately understand the range of strong seismic activity in Xinjiang, we first consider applying the LGCP model to describe complex earthquake sequences and further analyze the spatial distribution of seismicity in Xinjiang. Second, we will characterize the sequence parameters of moderate-to-strong earthquakes in Xinjiang from 2009 to 2023 in various regions and over time using the ETAS (Epidemic Type Aftershock Sequence Model) and INLA (Integrated Nested Laplace Algorithm). In conclusion, a daily retrospective analysis of earthquake forecasting following significant seismic events in Xinjiang is conducted. This analysis is useful in assessing the pattern of upcoming seismic activity.

## Research models and methods

### Epidemic Type Aftershock Sequence (ETAS) modeling

The temporal ETAS model is a labeled Hawkes process model [27] whose labeling variable is the magnitude size of the earthquake. The model consists of three components: a background rate term(a representation of earthquakes induced by past earthquake activity), a triggering event rate term, and a magnitude distribution independent of space and time. It is usually considered that the magnitude distribution of an event is independent of the spatial and temporal distributions, then the ETAS conditional density is usually expressed as the product of the Hawkes process model and the magnitude distribution $\pi(m)$, which is expressed in the following form:

$$\lambda_{ETAS}(t, m | H_t) = \lambda_{Hawkes}(t | H_t)\pi(m) \tag{1}$$

In seismology, the magnitude distribution $\pi(m)$ for is usually the G-R distribution [28], and the G-R distribution (magnitude-frequency distribution) is commonly of the form: $\lg N(m) = a - bm$, which the exponent b is usually near 1 [29]. Focus will be placed on the Hawkes part of the model, which is the conditional density of the Hawkes process for a given historical process

$H_t$:

$$\lambda_{Hawkes}(t|H_t) = \mu + \sum_{(t_h, m_h) \in H_t} K e^{\alpha(m_h - M_0)} \left( \frac{t - t_h}{c} + 1 \right)^{-p} \tag{2}$$

In the formula, the time evolution of earthquakes is influenced by five parameters: $\mu$, $K$, $\alpha$, $c \geq 0$ and $p \geq 1$. The parameter $\mu$ represents the background seismicity rate, $p$ denotes how fast or slow the aftershock sequence decays, $c$ denotes the time for when the frequency of aftershocks peaks after the mainshock, $K$ denotes the degree of activity of the aftershock, $\alpha$ denotes the ability to trigger a secondary aftershock, and $M_0$ is the minimum completeness magnitude such that $m_h \geq M_0$.

In this paper, we will use the INLA algorithm as a tool for fitting this model, which entails decomposing the log-likelihood of the Hawkes process into multiple parts and then returning the exact log-likelihood function as the sum of these parts. The general idea is to use the Integrated Nested Laplace Approximation (INLA) method to infer the model parameters by approximating individual components linearly to the posterior. The INLAbru package provides both the linearization and the posterior mode query internally. (This is an open-source package builds on the R-INLA package to provide suggested bayesian inference methods for point, count, and geographic sample data using integrated nested laplace approximations. Apply INLA to a variety of issues, including earthquake forecasting [6, 7], at the following URL: https://github.com/inlabru-org/inlabru.) We only need to provide the Hawkes log-likelihood function:

$$L(\theta|H) = -\Lambda(T_1, T_2) + \sum_{(t_i, m_i) \in H} \log \lambda(t_i | H_{t_i})$$

$$H = \{(t_i, m_i) : t_i \in [T_1, T_2], m_i \in (M_0, )\} \tag{3}$$

$$\Lambda(T_1, T_2) = \int_{T_1}^{T_2} \lambda(t|H_t) dt$$

$$= (T_2 - T_1)\mu + \sum_{(t_i, m_i) \in H} \int_{T_1}^{T_2} K e^{\alpha(m_i - M_0)} \left( \frac{t - t_i}{c} + 1 \right)^{-p} dt$$

$$= (T_2 - T_1)\mu + \sum_{(t_i, m_i) \in H} K e^{\alpha(m_i - M_0)} \int_{T_1}^{T_2} \left( \frac{t - t_i}{c} + 1 \right)^{-p} dt \tag{4}$$

$$= (T_2 - T_1)\mu + \sum_{(t_i, m_i) \in H} K e^{\alpha(m_i - M_0)} \frac{c}{p-1} \left( \left( \frac{\max(t_i, T_1) - t_i}{c} + 1 \right)^{1-p} - \left( \frac{T_2 - t_i}{c} + 1 \right)^{1-p} \right)$$

$$= \Lambda_0(T_1, T_2) + \sum_{(t_i, m_i) \in H} \Lambda_i(T_1, T_2)$$

The above integral can be understood as the sum $\Lambda_i(T_1, T_2)$ of the number of background events $\Lambda_0(T_1, T_2)$ and the number of triggered events ti per event. The method requires approximating the integral linearized functions $\Lambda_0(T_1, T_2)$ and $\Lambda_i(T_1, T_2)$, and in order to further improve the accuracy of the approximation, for each integral $\Lambda_i(T_1, T_2)$, we consider a further partition of the integration interval $[\max(T_1, t_i), T_2]$ into $B_i$ time boxes

$(t_0^{(b_i)}, \ldots, t_{B_i}^{(b_i)}; t_0^{(b_i)} = \max(T_1, t_i); t_{B_i}^{(b_i)} = T_2)$, so that the integral becomes:

$$\Lambda(T_1, T_2) = \Lambda_0(T_1, T_2) + \sum_{(t_i, m_i) \in H} \sum_{j=0}^{B_i - 1} \Lambda_i(t_j^{(b_i)}, t_{j+1}^{(b_i)}) \tag{5}$$

The Hawkes log-likelihood function is then expressed in the following form:

$$L(\theta | H) = -\Lambda_0(T_1, T_2) - \sum_{(t_i, m_i) \in H} \sum_{j=0}^{B_i - 1} \Lambda_i(t_j^{(b_i)}, t_{j+1}^{(b_i)}) + \sum_{(t_i, m_i) \in H} \log \lambda(t_i | H_{t_i}) \tag{6}$$

## Based on the INLA algorithm

The following section outlines a specific methodology for implementing the ETAS model based on the Bayesian INLA algorithm and utilizing the R-Inlabru package. The technical method is very different from earlier ETAS model implementations. Specifically, no clustering algorithm is used in the computational approach to allocate observations to the triggering component of the background rate or intensity. Moreover, the user does not need to explicitly program the algorithm but only needs to provide the approximation function for the three parts of the log-likelihood and build the different log-likelihood components, making it an effective alternative to the Bayesian-based approach of existing ETAS models.

The INLAbru package is built on the R-INLA package and can provide easier Bayesian inference methods for spatial point processes, counting, gridding, and geographic sampling [30]. We will use this package to build and run the following models to fit actual observations using the LGCP model, which can accommodate data including points, counts, geographic samples, or distance sample data, and which provides methods for fitting spatial density surfaces and estimating abundance, as well as for mapping and forecasting. Similarly, we will decompose the log-likelihood function of the Hawkes process into multiple parts implemented separately using the INLAbru-based method proposed by Naylor et al (2023) [31], linearly approximating each single component and applying the Integrated Nested Laplace Approximation (INLA) method to infer the model parameters, a brief description of which is developed below.

We will fit the above ETAS model using the INLA algorithm, whose approximate form for the log-likelihood function has been given in the section above:

$$\tilde{L}(\theta, \theta^*) = -\tilde{\Lambda}_0(\theta, \theta^*) - \sum_{h=1}^{n} \sum_{i=1}^{B} \tilde{\Lambda}_h(b_i, \theta, \theta^*) + \tilde{S}L(h, \theta, \theta^*) \tag{7}$$

The method combines three Poisson model implementations on different datasets in the INLAbru package, where INLA is referenced to implement the Poisson model only for computational efficiency. Specifically, the internal log-likelihood used by INLA for the Poisson model will be utilized to obtain an approximation of the log-likelihood of the ETAS model.

The generic Poisson model (which is located at $x_i$, counts as $c_i$, and exposures as $E_1, \ldots, E_n$) in the INLAbru package will have its log-likelihood function expressed as:

$$L_P(\theta) \propto -\sum_{i=1}^{n} \exp\{\bar{f}(x_i, \theta, \theta^*)\} * E_i + \sum_{i=1}^{n} \bar{f}(x_i, \theta, \theta^*) * c_i \tag{8}$$

An alternative Poisson model is used for each log-likelihood component, whose log-likelihood is given by the above equation, with appropriate counts and exposures chosen. Table 1 gives

**Table 1. Approximations of the log-likelihood for each of the three components and their alternative Poisson models.**

| Part | Object | Approximate | Surrogate log $\lambda_P$ | Number of data points | Counts and Exposures |
|---|---|---|---|---|---|
| Part 1 | $\Lambda_0(x)$ | $\exp\{\log \Lambda_0(\theta, \theta^*)\}$ | 1 | 1 | $c_i = 0, e_i = 1$ |
| Part 2 | $\sum_{i=1}^{n} \sum_{i=1}^{B_h} \Lambda_h(b_i, h)$ | $\sum_{i=1}^{n} \sum_{i=1}^{B_h} \exp \log \Lambda_h(b_i, h)$ | $\log \Lambda_0(b_i, h)$ | $\sum_{h} B_h$ | $c_i = 0, e_i = 1$ |
| Part 3 | $\sum_{h=1}^{n} \log \lambda(x_h)$ | $\sum_{h=1}^{n} \exp \log \lambda(x_h)$ | $\log \lambda(x)$ | n | $c_i = 1, e_i = 0$ |

an approximation of the three components of the log-likelihood function and the alternative Poisson model it uses to represent it.

We only need to create datasets with counts $c_i$, exposures $e_i$, and information about events $x_i$, where this information represents the different log-likelihood components, and provide the functions $\log \Lambda_0(x)$, $\log \Lambda_0(b_i, h)$, and $\log \lambda(x)$, and the linearization, as well as the retrieval of the posterior distributions of its parameters, are performed automatically from within INLAbru.

It is worth noting that the INLA method is designed for latent Gaussian models so that all parameters in the fitted model should obey a normal distribution. However, this does not hold for some ETAS model parameters, and to overcome this problem a transformation based on a probabilistic integral transform will be used: Given a continuous random variable X with cumulative distribution function (CDF) FX(-) which is uniformly distributed in (0,1):

$$Y = F_X(X)$$
$$Y \sim Unif(0, 1), X = F_X^{-1}(Y)$$

$$(9)$$

Briefly, the approach is to treat each parameter as having a standard normal distribution and then convert it to the target distribution. Suppose that $\theta$ has the starting distribution of the CDF $F_\theta()$ and that we wish to convert it to $\eta(\theta)$ has the target CDF FY(-), converted to:

$$\eta(\theta) = F_X^{-1}(F_\theta(\theta)) \tag{10}$$

We can consider a set of constraint parameters $\theta_\mu, \theta_K, \theta_\alpha, \theta_c$ and $\theta_p$ having standard normal prior distributions, representing $\mu, K, \alpha, c$ and $p$, respectively, and then convert them to the desired prior distributions. Using the above approach, different priors can be considered for implementation, while the choice of different priors can also affect the convergence ability of the algorithm.

## Research data

Earthquake activity in Xinjiang is characterized by high intensity, high frequency, and wide distribution, which makes it a major area of strong seismic activity in mainland China. For this reason, the Xinjiang region will be the focus of this paper's investigation. The geographic location of the selected study region is between $73°40'E$ and $96°18'E$, and between $34°25'N$ and $48°10'N$. Xinjiang is located in the front zone of the collision between the Indian Ocean Plate and the Asian-European Plate, and there are five major seismic zones, which are the Altai seismic zone, the North Tianshan seismic zone, the South Tianshan seismic zone, the West Kunlun Mountain seismic zone, and the Altun Mountain seismic zone, respectively, from north to south. According to the statistics of China Seismological Network, since 2009, a total of 2905 earthquakes have occurred in Xinjiang region, of which 12 earthquakes reached magnitude 6 and above, with the largest magnitude reaching 7.3 (Hotan, February 12, 2012).

The earthquake catalog used in this study is provided by the National Seismic Cataloging System (NSCS) from January 1, 2009, to July 2, 2023, by the China Earthquake Network Center (CENC). At the same time, assessing the completeness of the earthquake catalog magnitude is an essential step in any earthquake activity analysis. The completeness magnitude of an earthquake catalog is defined as the smallest magnitude at which 100 percent of an earthquake is detected within a spatiotemporal volume. Its correct estimation is critical because too high a value can lead to under-sampling and thus discard available data, while too low a value can lead to erroneous values of seismic activity parameters and thus biased analysis with incomplete data. In this paper, we will use the maximum curvature method (MAXC) [32] is a fast and direct method to estimate the completeness magnitude of an earthquake. It is defined by calculating the maximum of the first derivative of the cumulative magnitude frequency distribution curve, which usually matches the corresponding magnitude of the earthquake with the highest frequency in the non-cumulative magnitude frequency distribution. The magnitude frequency distribution, which can effectively show the probability density distributions of earthquakes with different magnitudes and the cumulative probability density distributions of earthquakes with different magnitude lower bounds, is the most easily comprehensible image for earthquake catalogs. The magnitude time distribution describes the magnitude over time, and the magnitude frequency distribution describes the G-R relationship [33]. The magnitude frequency distribution, which can effectively show the probability density distributions of earthquakes with different magnitudes and the cumulative probability density distributions of earthquakes with different magnitude lower bounds, is the most easily comprehensible image for earthquake catalogs. The magnitude time distribution describes the magnitude over time, and the magnitude frequency distribution describes the G-R relationship.

$$\log_{10}N = a - b(M - M_c) \tag{11}$$

where N is the number of earthquakes of magnitude Mc or higher, the a value is the yield of the earthquake, and the b value describes the relative proportion of the number of earthquakes of different magnitude sizes in the earthquake catalog. From the cumulative frequency distribution (Cum. FMD) and non-cumulative frequency distribution (Non Cum. FMD) in Fig 1, it can be seen directly that the completeness magnitude is 3.0.

## Descriptive analysis of earthquake sequences in Xinjiang region based on the LGCP model

In this paper, the earthquake sequences occurring in Xinjiang region during 2009-2023 will be described based on the LGCP model, and the constructed model is as follows:

$$\log(u(s)) = \beta_0 + \zeta(s) + \varepsilon \tag{12}$$

In Eq 13, $\beta_0$ is the intercept term; $\varepsilon$ is the error term; and the spatially varying Gaussian random field $\zeta(s)$ explains the spatial variations in the model that are not explained by the deterministic component. In this way, the Gaussian random field models the spatial structure by explaining the spatial correlation between observations, where the Gaussian field is specifically defined as follows:

$$\zeta(s) \sim GaussianField(0, \Sigma) \tag{13}$$

Its mean is 0 and variance is $\Sigma$. The calculation of the covariance is more complicated, instead of calculating all the values independently, it is better to use the standard correlation function to describe the correlation between points, using the Matern correlation function [34] can be

## Frequency−Magnitude Distribution

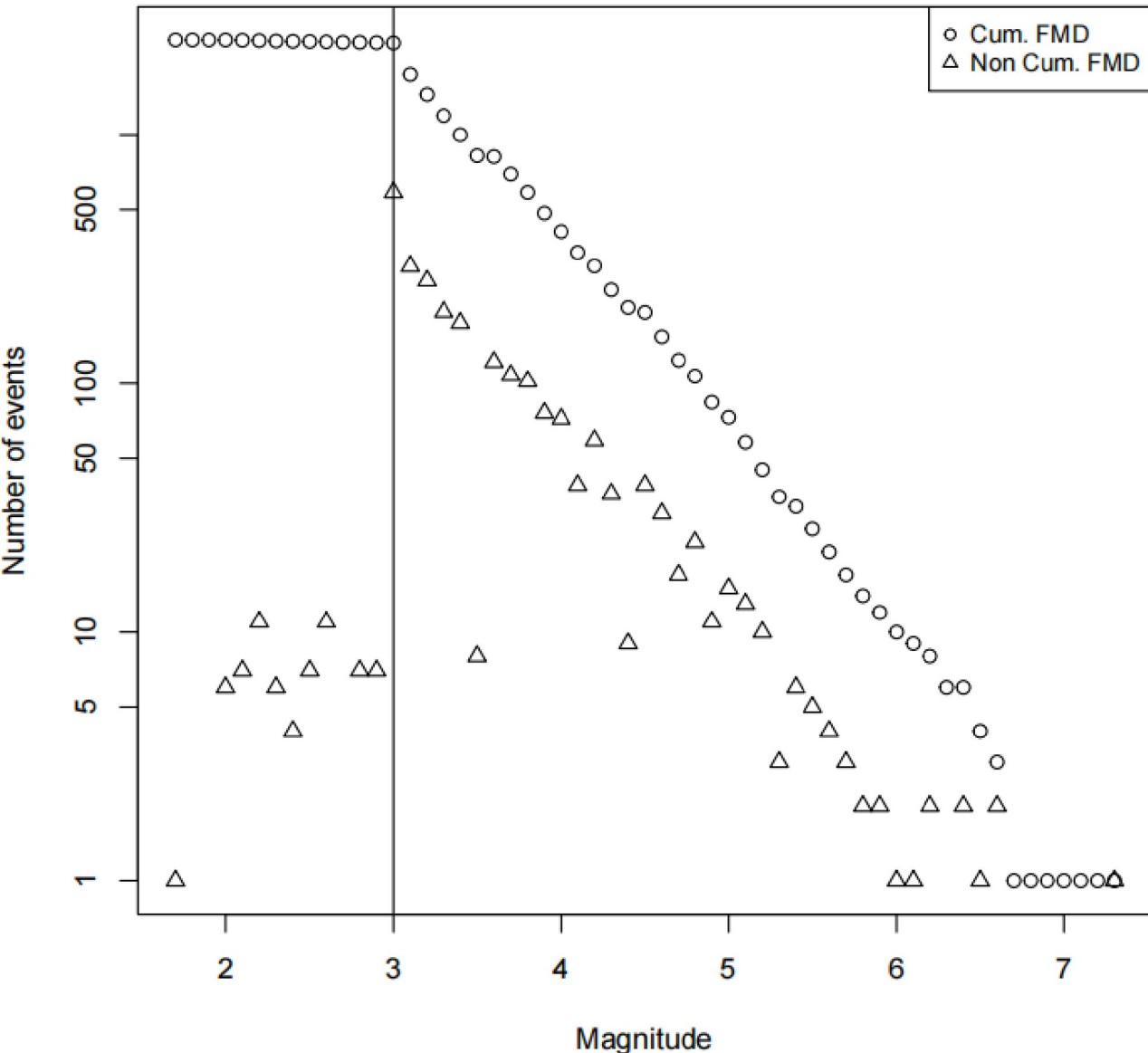

**Fig 1. The G-R relation distribution (FMD) of the earthquake catalog in Xinjiang region during 2009-2023, with triangles and circles representing non-cumulative and cumulative frequency distributions.**

used to define the covariance, which is denoted as:

$$\Sigma = Cov_M = \sigma^2 Corr_M \tag{14}$$

where $\sigma^2$ is the variance parameter and CovM and CorrM are the Matern covariance and correlation functions, respectively. The Matern correlation function is specified as [35].

$$Corr_M = \frac{2^{1-v}}{\Gamma(v)}\left(\sqrt{2v}\frac{d}{0.5r}\right)^v K_v\left(\sqrt{2v}\frac{d}{0.5r}\right) \tag{15}$$

In the formula, $d$ is the distance between two observation points, $\Gamma(x)$ and $K_v$ represent the gamma function and Bessel function, $v$, $r$ and $\sigma$ are the smoothing, range, and standard deviation parameters of the random field, respectively. In this paper, we can describe the Gaussian random field by the range as well as the standard deviation parameter, so we need to set the PC prior [36] so that $\Pr[r < 0.01 = 0.01]$, which is to avoid the value of the range is too small, which will lead to overfitting; meanwhile, $\Pr[\sigma > 0.1] = 0.01$, which is to prevent the standard deviation from being too high, and these parameters need to be adjusted according to the actual problem.

## LGCP model fitting results and analysis

We consider all earthquakes occurring in the Xinjiang region from 2009 to 2023 as response variables, which are fitted based on the logarithmic Gaussian Cox process model (LGCP) and using the INLA algorithm.

First, the parameter estimates (range, standard deviation) of the intercept term as well as the Gaussian random field in the LGCP model can be obtained, as shown in Table 2. Among them, the mean value of the standard deviation parameter is 1.121, and its confidence interval is (0.459, 2.763), since zero is not included in the interval can indicate that the earthquakes in Xinjiang region have spatial aggregation. In addition, the fitted mean value of the range parameter will correspond to a smaller spatial autocorrelation value, and its fitted mean value of 1.995 can indicate that the occurrence of any earthquake will have a smaller effect on the occurrence of other earthquakes with latitude/longitude exceeding 1.995 degrees. Secondly, the spatial distribution pattern of earthquakes can be obtained through the posterior mean results of spatial random fields of LGCP model. Only the posterior fitting results of 10 regions in Xinjiang are listed here, as shown in Table 3. Table 3 is arranged from top to bottom according to the posterior mean from largest to smallest. The higher the posterior mean is, the higher rate of earthquakes in the region. On the contrary, it means that no earthquake has been recorded in the region or few earthquakes have been recorded in the region. It can be clearly

**Table 2. LGCP model: Range and standard deviation parameter fitting results for intercept term and random field.**

| LGCP model parameter | Mean | Variance | 0.025quant | 0.975quant |
|---|---|---|---|---|
| Range | 1.995 | 0.096 | 0.208 | 2.424 |
| Standard deviation | 1.121 | 0.053 | 0.459 | 2.763 |
| Intercept | -7.664 | 0.257 | -8.169 | -7.160 |

**Table 3. LGCP model: Spatial random field posterior mean fitting results of 10 regions in Xinjiang.**

| Region | Mean | Variance | 0.025quant | 0.975quant |
|---|---|---|---|---|
| Kizilsut | 0.007178 | 0.001486 | 0.004911 | 0.010854 |
| Kashgar | 0.002697 | 0.000753 | 0.001549 | 0.004441 |
| Bortala | 0.002160 | 0.000741 | 0.001059 | 0.004242 |
| Hotan | 0.000562 | 0.000262 | 0.000201 | 0.001208 |
| Yili | 0.000390 | 0.000676 | 0.001358 | 0.003935 |
| Bayingolin | 0.000345 | 0.000196 | 0.000115 | 0.000833 |
| Tacheng | 0.000325 | 0.000211 | 0.000100 | 0.000875 |
| Changji | 0.000296 | 0.000186 | 0.000084 | 0.000788 |
| Kami | 0.000206 | 0.000147 | 0.000048 | 0.000583 |
| Altay | 0.000198 | 0.000140 | 0.000046 | 0.000569 |

seen from Table 3 that there are significant differences in the spatial distribution of earthquakes in Xinjiang. The posterior mean of Hotan, Kashgar, Kizilsut, Yili, and Bortala is higher than that of other regions, indicating that these regions have the highest frequency of corresponding earthquakes.

In this section, the LGCP model is used to study the spatial distribution characteristics of earthquakes in Xinjiang. The following focuses on the top five regions ranked by the spatial random field posterior mean in Table 3 and the regions with high seismicity such as Hotan, Kashgar, Kizilsut, Yili and Boltala.

## Research results and analysis

### Results of fitting ETAS model based on INLA and goodness-of-fit test

**Fitting results of ETAS model parameters in different regions of Xinjiang.** The earthquake sequences occurring in the following five regions were selected through the combination of latitude-time, longitude-time, and epicenter distribution plots: the 2012 Hotan region Ms6.0 (which includes the August 2012 Ms6.2, the February 2014 Ms7.3, the July 2015 Ms6.5, and the June 2020 Ms6.4), the 2016 Kizilsut region Ms6.7, 2012 Yili region Ms6.0 (which contains November 2012 Ms6.6), 2020 Kashgar region Ms6.4, 2017 Bortala region Ms6.6, and brief information on the occurrence of the mainshock for the selected earthquake sequences of the five regions ($M_0$=3.0) is shown in Table 4 below (containing coordinate information, mainshock magnitude, and time of occurrence).

In this section, the model parameters of earthquake sequence activity in various regions of Xinjiang will be studied using the ETAS model. We compare the relative magnitudes of model parameters that are physically significant and play a controlling role and discuss physical issues such as the rate of attenuation of seismic sequence activity rates and aftershock stimulation in various regions.

Table 5 gives the fitting results of the ETAS model parameters after the occurrence of the mainshock in these five selected regions. We found that there are obvious regional differences in the fitting results of the model parameters of the earthquake sequences located in different regions of Xinjiang, which may be due to the different magnitude sizes of the mainshock, different types of faults in each region, and different sequence types of aftershocks. The rate of decay of the aftershock sequences and the ability to stimulate aftershocks can be characterized by the model parameters of the sequences. Comparing the results of fitting the ETAS model parameters of the earthquake sequences in different regions, the following sequence characteristics can be obtained: (1) in terms of the ability of the mainshock to stimulate aftershocks and

**Table 4. Summary information on the occurrence of mainshocks of selected earthquake sequences in five regions of Xinjiang.**

| Region | Latitude(N) | Longitude(E) | Mainshock magnitude(Ms) | Mainshock date(dd/mm/yyyy hh:mm:ss) |
|---|---|---|---|---|
| Hotan | 39.4 | 81.3 | 6.0 | 09/03/2012 06:50:09 |
| | 35.9 | 82.5 | 6.2 | 12/08/2012 18:47:12 |
| | 36.1 | 82.5 | 7.3 | 12/02/2014 17:19:50 |
| | 37.6 | 78.2 | 6.5 | 03/07/2015 09:07:46 |
| | 35.7 | 82.3 | 6.4 | 26/06/2020 05:05:20 |
| Yili | 43.6 | 82.4 | 6.0 | 01/11/2011 08:21:28 |
| | 43.4 | 84.8 | 6.6 | 30/06/2012 05:07:31 |
| Kizilsut | 39.3 | 74.04 | 6.7 | 25/11/2016 22:24:30 |
| Bortala | 44.3 | 82.9 | 6.6 | 09/08/2017 07:27:52 |
| Kashgar | 39.83 | 77.21 | 6.4 | 19/01/2020 21:27:55 |

**Table 5. The fitting values and 95% confidence interval results of the ETAS model after the mainshock of the selected earthquake sequences in five regions.**

| Region | $\mu$ | $K$ | $\alpha$ | $c$ | $p$ |
|---|---|---|---|---|---|
| Hotan | 0.051(0.048 ± 0.054) | 0.176(0.158 ± 0.192) | 1.292(1.246 ± 1.338) | 0.032(0.024 ± 0.037) | 1.332(1.307 ± 1.354) |
| Yili | 0.026(0.024 ± 0.028) | 0.001(0.000 ± 0.001) | 5.975(4.540 ± 6.711) | 0.889(0.341 ± 1.258) | 5.052(3.492 ± 6.864) |
| Kizilsut | 0.101(0.096 ± 0.106) | 0.045(0.036 ± 0.053) | 1.895(1.824 ± 1.965) | 0.047(0.032 ± 0.058) | 1.400(1.345 ± 1.443) |
| Bortala | 0.013(0.011 ± 0.014) | 0.002(0.000 ± 0.002) | 3.475(2.707 ± 3.833) | 0.029(0.013 ± 0.036) | 1.384(1.293 ± 1.448) |
| Kashgar | 0.062(0.056 ± 0.068) | 0.180(0.131 ± 0.222) | 0.618(0.443 ± 0.797) | 1.548(0.137 ± 2.145) | 4.749(3.309 ± 5.200) |

aftershocks to subsequent earthquakes (determined by the values of alpha), we can observe that the regional differences in alpha values are relatively significant, and if we rank the ability of these five regions to trigger secondary aftershocks according to the size of alpha values: Yili region > Bortala region > Kizilsut region > Hotan region > Kashgar region. In general, the aftershock-triggering ability of Yili region is significantly higher than that of the other Xinjiang four regions, and Kashgar region is the weakest. Moreover, we consider comparing the alpha values of Xinjiang regions with the mean values of alpha in the ETAS model parameters in the early post-earthquake phase of moderate-to-strong earthquake sequences in the Chinese mainland region, and the distribution of the mean values of alpha in the range of 1.049 ± 0.316 [37]. Compared with the alpha values in Table 5, it can be seen that except for Kashgar region, the alpha of other regions is higher than the average value of Chinese regions, and thus the ability to stimulate aftershocks is stronger; (2) the p-value characterizes the speed of the sequence rate of decay, if the p-value is large, the sequence rate of decay is fast, if the p-value is small, the sequence rate of decay is slow. It can be found from Table 5 that the regional differences in p-values are also relatively large, with Yili region having the fastest attenuation, Kashgar region the second, Hotan region the third, and Kashgar region the third. Kashgar region is the second fastest, and the lowest p-value in Hotan region indicates that the aftershock sequence in Hotan region, which is located in the southernmost part of Xinjiang, has a longer duration.

**Time-varying characteristics of earthquake sequence parameters.** Due to the different rates of tectonic movement of intraplate earthquakes, the duration of the earthquake sequence in each region is different. Therefore, the parameters of the ETAS model obtained by fitting will be different depending on the fitting cutoff time of the selected earthquake sequence. To investigate the variation characteristics of earthquake sequence parameters with time in the short term after the main earthquake, we take the Ms7.3 earthquake sequence in Hotan region on February 12, 2014, as an example. We consider that the length of time for the parameters to reach stability is affected under different earthquake completeness magnitudes, the earthquake completeness magnitude is fixed at 3.0, and the cut-off times Tend to be set as [1.0, 2.0, 3.0, . . ..., 30.0] and use INLA algorithm to estimate the parameters respectively. The changes of model parameters with the duration of the series were obtained as shown in Fig 2.

As can be seen in Fig 2, during the 15 days following the main earthquake, the model parameters changed as follows: first, the fitted value of $\mu$ in the figure changes significantly from 16.084 to 0.610, which is the most significant change among all the parameters, and reaches stability on the 10th day. Secondly, the fitted p-value gradually decreases from 6.515 to 2.355, which is also a more significant change and reaches stability on day 12. On the contrary, the fitted K-value hardly changes and stays very flat around 0.007, and the c-value also stays around 0.149, with relatively flat changes. Overall, there were differences in the degree of variation from the sequence parameter values, with the minimum degree of variation in the parameter c and K values, while the degree of variation in the parameter mu, p, and alpha values was very obvious. The reason for the above result is that the parameter c is a very small positive

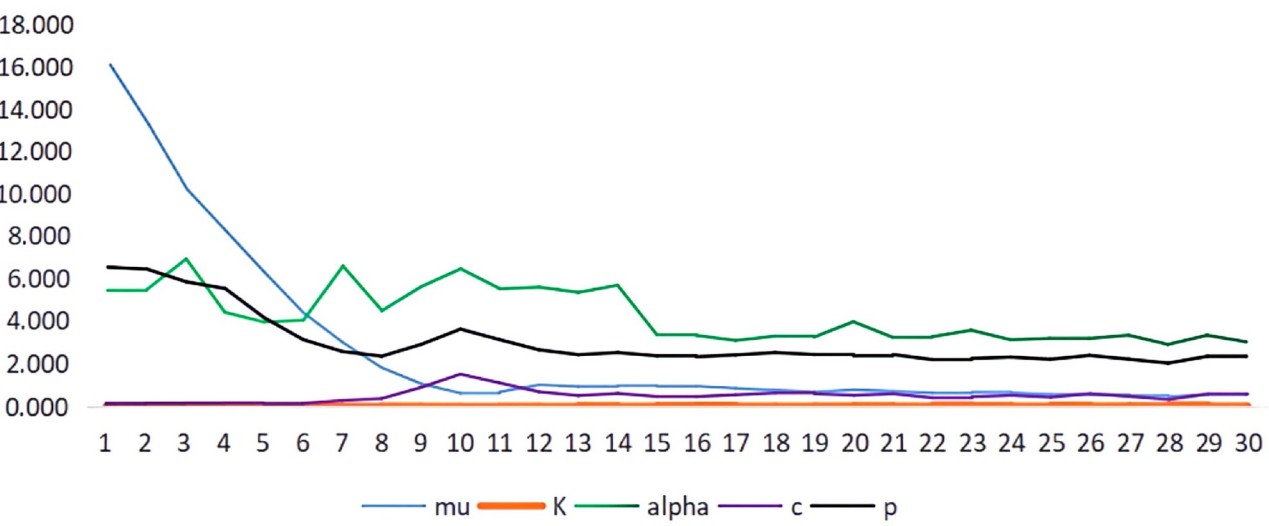

**Fig 2. Variation of ETAS model parameters with cutoff time.**

constant, which is mainly considered to include a small constant term in the model equation to avoid a zero denominator. It therefore does not have a very clear physical meaning. It can also be seen from Eq 2 that the parameters p and alpha are in exponential positions in the model expression, and thus c has less influence compared to p and alpha. Therefore, among the parameters of the ETAS model, the variation of c over time is relatively smooth, while the variation of alpha and p is more obvious. It is worth noting that the fitted alpha values have obvious irregular ups and downs, with sudden increases and decreases over the 15 d period, with the largest change of 2.52. For the changes of the above ETAS model parameters with the sequence fitting cutoff time, the alpha and p values both show sudden jumps in the 14 days after the earthquake, especially the alpha value of the ability to stimulate the secondary after-shocks has the most drastic change, and the p-value has a certain magnitude of fluctuation in the 13 days after the earthquake. The reason for the "sudden" changes in the sequence parameters can be analyzed from the earthquake sequence activity, which may be due to the occurrence of aftershocks of larger magnitude at the early stage after the main earthquake, thus the earthquake sequence has a large change in the decay rates of aftershock and the excitation degree of aftershock. However, this kind of sequence activity is only a representation analysis, and the adjustment of the stress field in the physical source area or the adjustment and change of the aftershock rupture mechanism needs to be further studied.

## Model fit goodness-of-fit test

In the following, we take the results of fitting the 2014 Ms7.3 earthquake sequence in Hotan as an example and compare the results of implementing this model using INLAbru with the results obtained using the R-bayesianETAS package (Ross 2021) [38] (the R-bayesianETAS package provides an MCMC implementation of the ETAS model), this section is intended to demonstrate the advantages of this computational method as well as to test the model's goodness-of-fit.

Here we consider the use of the stochastic time-varying theorem as a method of the accuracy of the results of the method fitting the ETAS model. The stochastic time-varying theorem [39], assumes that in time [0, T], $H = \{t_1, \ldots, t_k\}$ is a point process with conditional intensity $\lambda$

$(t|H)$, which is expressed as follows:

$$\Lambda(t_i|H) = \int_{M_0}^{\infty} \int_0^t \lambda(t, m|H)dtdm \tag{16}$$

In other words, the quantity $\Lambda(t)$ can be regarded as the expected number of points in the time domain, and we obtain the sequence $\Lambda(t_1), \ldots, \Lambda(t_n)$ should be associated with the actual observation points located at $t_1, \ldots, t_n$ the cumulative number of actual observation points. Below, Fig 3 presents the sequence values generated by the two implementations, $\Lambda_{mcmc}(t_1)$, $\ldots, \Lambda_{mcmc}(t_n)$ (blue line), $\Lambda_{Inlabru}(t_1), \ldots, \Lambda_{Inlabru}(t_n)$ (green line) and with the actual cumulative number $N(t_1), \ldots, N(t_n)$ (black dots) was compared.

Fig 3 (left) shows the fitted ETAS model curves and the cumulative number of earthquakes for the Ms6.0 earthquake sequence in Hotan using the Inlabru and MCMC methods. We found that the results of fitting the ETAS model based on the INLAbru and MCMC methods are nearly the same and the generated series values are very close to the actual values, which indicates that the model results obtained by fitting the two methods reflect the data of the region very well. In addition, we can plot $\Lambda_{Inlabru}(t_i)$, $\Lambda_{MCMC}(t_i)$ as in Fig 3(right) the same goodness-of-fit test can be carried out, the principle is to fit the obtained model to the actual data the better the line is closer to the black dashed straight line (the more similar to the theoretical straight line of the unit-rate Poisson process). Therefore, a conclusion consistent with Fig 3 (left) can be obtained.

In the following, we first consider comparing the occurrence time and magnitude size of the earthquake sequences generated by fitting the ETAS model based on the INLAbru and MCMC methods. It is evident from Fig 4 that the earthquake sequences produced by the INLAbru and MCMC methods differ significantly in terms of both the magnitude and timing

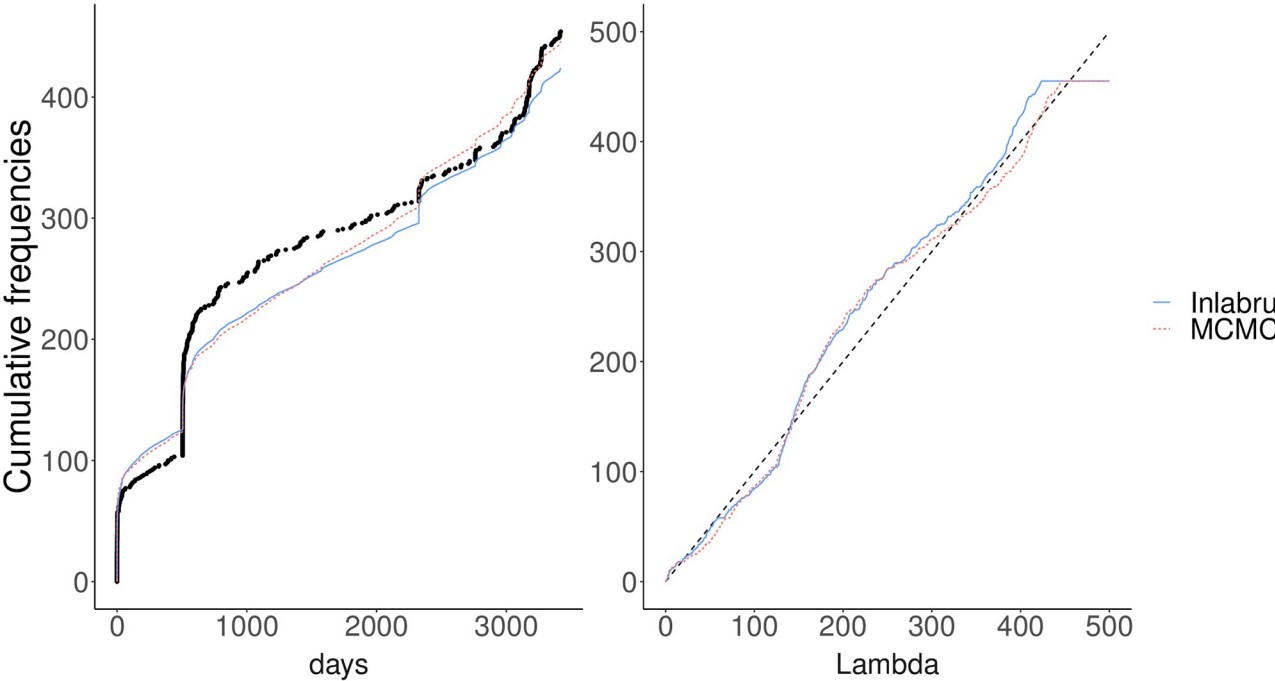

**Fig 3. Cumulative number of earthquakes for the Ms6.0 earthquake sequence in the Hotan region versus using Inlabru and MCMC to realize the ETAS model fitting curve comparison.**

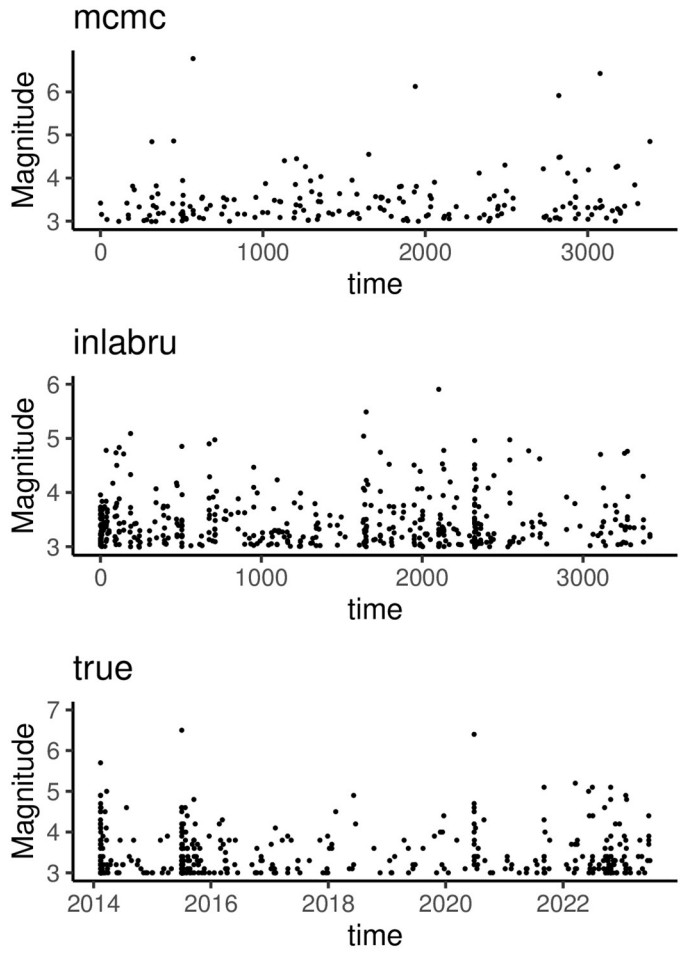

**Fig 4. Time-magnitude plots of earthquake sequences generated by the ETAS model and actual earthquake sequences are fitted based on the INLAbru, MCMC method.**

of the earthquakes. Following comparing the two methods, we can say that the earthquake sequences generated by fitting the ETAS model using the INLAbru method have occurrence times and magnitude sizes that are closer to the actual earthquakes. Second, the MCMC method takes 1.48207 minutes to compute when dealing with the same number of earthquake sequences (using the Hotan region's 2014 Ms7.3 earthquake sequence, which consists of 455 events as an example). In contrast, the INLAbru method, which relies on the approximate parameter a posteriori technique, takes only 0.29984 minutes to realize the model fitting. The INLAbru technique has the benefit of drastically cutting calculation time, which becomes more noticeable when dealing with a sizable dataset of earthquakes and numerous variables.

## Analysis of retrospective forecast results

Here, we take the 2014 Ms7.3 earthquake sequence in Hotan region as an example to conduct a retrospective daily forecasting experiment. Specifically, 1000 a posteriori samples of the ETAS model parameters of the earthquake sequences in the region are taken as "input samples" for the forecasting and are used to generate 1000 synthetic catalogs of earthquakes (one for each sequence) starting from February 12, 2014, which is the same period as that of the

Ms7.3 sequence in Hotan region. ETAS model parameter a posteriori sample is used to generate one earthquake synthetic catalog. For each forecast period defined $(t_j, t_{j+1})$, it is assumed that all earthquakes are known to occur strictly before the forecast period, i.e., $H_{tj}$. where we will refer to the Earthquake Predictability (CSEP, 2020) in which it is assumed that if an earthquake with a magnitude greater than 5.5 occurs during the forecast period $(t_j, t_{j+1})$ at a recording time of $t_m$: $t_j < t_m < t_{j+1}$, then $t_j$, the $t_m$ period and start a new daily forecast from $t_m + dt$ ($dt > 0$).

First, we will generate the earthquake sequence within 100 days after the Feb. 12, 2014, Ms7.3 mainshock based on the ETAS parameters and obtain the quantitative value of the number of forecast earthquakes per day. The outcomes of the retrospective forecasting test are displayed in Fig 5. The number of observed earthquakes per forecast period is represented by the black dots in the figure, the median number of earthquakes in the synthetic catalog for each forecast period is shown by the red solid line, and the 95% forecast interval for the number of earthquakes in the synthetic catalog for each forecast period is shown by the orange shaded area. Overall, we can observe that almost all the observed numbers of earthquakes are included in the confidence intervals. The earthquake sequence tends to stabilize at the late stage of a strong earthquake, and we focus our attention on the short period after the Ms7.3 mainshock occurs in Hotan, which corresponds to the peak region of the red line in the figure. Table 6 gives the forecast and actual number of earthquakes in the five days after the mainshock, and

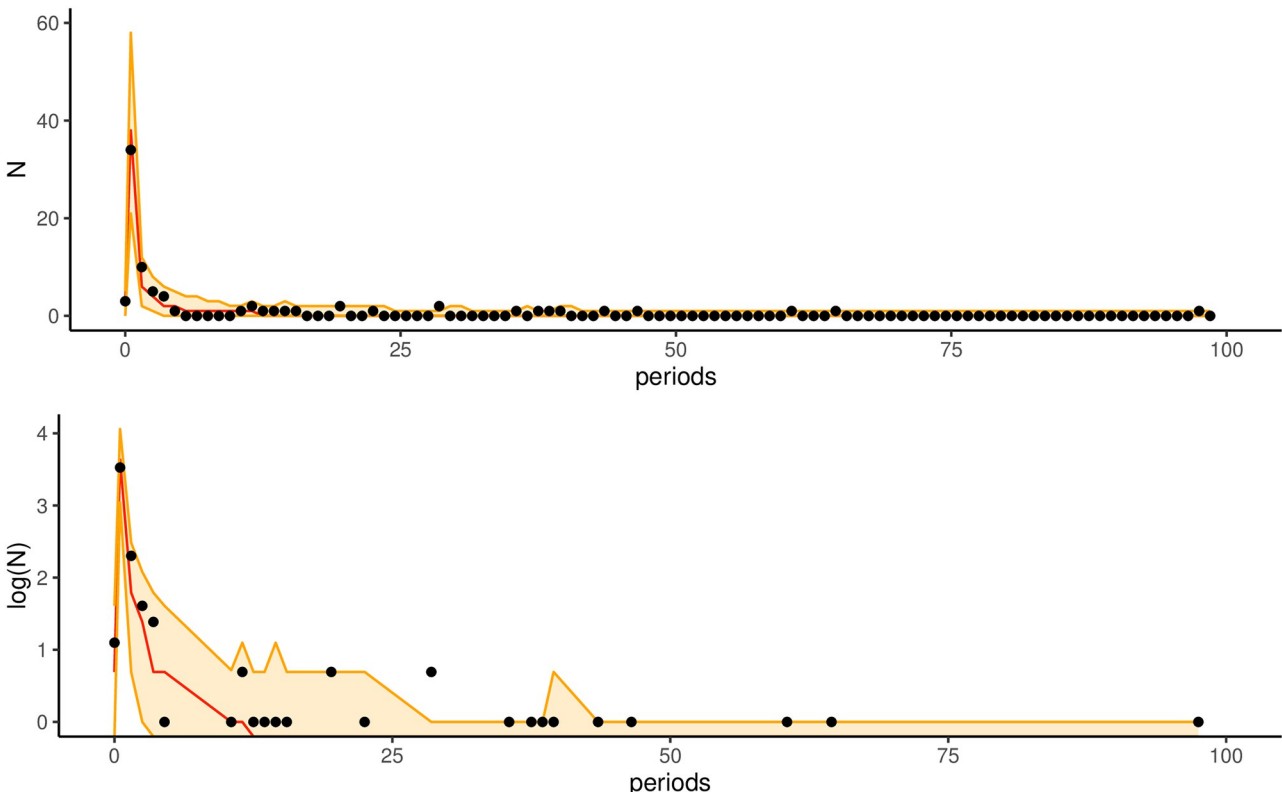

**Fig 5. Retrospective forecasting test (using the earthquake sequence from February 12, 2014, in Hotan region as an example).** Fig 5 (top), the black dots indicate the number of earthquakes observed in each forecast period, the red solid line indicates the median number of earthquakes in the synthetic catalog for each forecast period, and the orange areas indicate the 95% forecast intervals for the number of earthquakes in each time. The extreme values for each interval are the 2.5% and 97.5% quantiles of the number of earthquakes in the synthetic catalog that make up the daily forecast. Fig 5 (below) gives the logarithm of the number of earthquakes, thus enlarging the almost empty part of the value.

**Table 6. Intervals of the forecasted number of earthquakes and the actual number of earthquakes within five days after the mainshock of Ms7.3(February 12, 2014)in Hotan region.**

| Days | Lower(5%) | Median(50%) | Upper(95%) | True |
|------|-----------|-------------|------------|------|
| 1 | 0 | 2 | 5 | 3 |
| 2 | 21 | 38 | 58 | 34 |
| 3 | 2 | 6 | 12 | 10 |
| 4 | 1 | 4 | 8 | 5 |
| 5 | 0 | 2 | 6 | 4 |

the table shows that the mean value of the forecast number of earthquakes is very close to the actual value, which indicates that the model can forecast the number of early earthquakes in the forecast period accurately after the occurrence of the strong earthquake Ms7.3 (Hotan, February 12, 2014).

Second, we use the fitted mean values of the ETAS model parameters obtained from the fitting of the MCMC and INLA algorithms in simulation experiments to calculate the likely number of long-term earthquakes after the mainshock and to make comparisons of the outcomes of these two independent methods. We select the period of Ms7.3 occurrence in Hotan region since Feb. 12, 2014, as the starting time, and July 2, 2023, as the cutoff date (a total of 455 earthquakes occurred) to synthesize the 1000 earthquake catalogs. We chose the earthquake sequences from this region and time because of the large magnitude of the earthquakes and the long duration of the sequences, which therefore contain a large number of more complete earthquakes. At the same time, we select several historically strong earthquake earthquakes occurring during this period as fixed event points in each simulation experiment (in addition to Ms7.3 also includes two earthquakes, Ms6.5 on July 3, 2015, and Ms6.4 on June 26, 2020). Fig 6 extracts the earthquake catalogs corresponding to the 0.025, 0.5, and 0.975 quartiles in the distribution of the number of earthquakes used for the synthetic earthquake catalogs using the results of the two algorithms' fits, respectively, and plots the histograms of the forecast number of earthquakes for each month as well as those from the actual earthquake catalogs. The expected number of earthquakes obtained by using the INLA algorithm is closer to the observed number of earthquakes than those for the MCMC method. Based on the actual monthly event rate plotted in Fig 6, we can see that the INLA algorithm can more accurately forecast the number of earthquakes (per month) in the period of sudden (high) earthquakes compared with the MCMC algorithm and hence may be a more effective method of aftershock forecasting.

## Conclusion

To systematically investigate the ETAS model of earthquake sequences in Xinjiang, the spatial and temporal characterization of the parameters of earthquake sequences in Xinjiang in the National Seismic Cataloging System (NSCS) provided by the China Earthquake Network Center (CENC) from Jan. 1, 2009, to Jul. 2, 2023, and the forecasting of strong aftershocks are selected for the study.

Firstly, the spatial distribution characteristics of earthquakes are described based on the log-Gaussian Cox process model (LGCP), and the parameters of Gaussian random fields in the LGCP model are calculated by the INLA algorithm. The results show that the confidence interval of the random field standard deviation parameter does not contain zero, which indicates that the earthquakes in Xinjiang have spatial local effects. Then, according to the posterior mean of spatial random fields in LGCP model, it is found that the high risk areas of

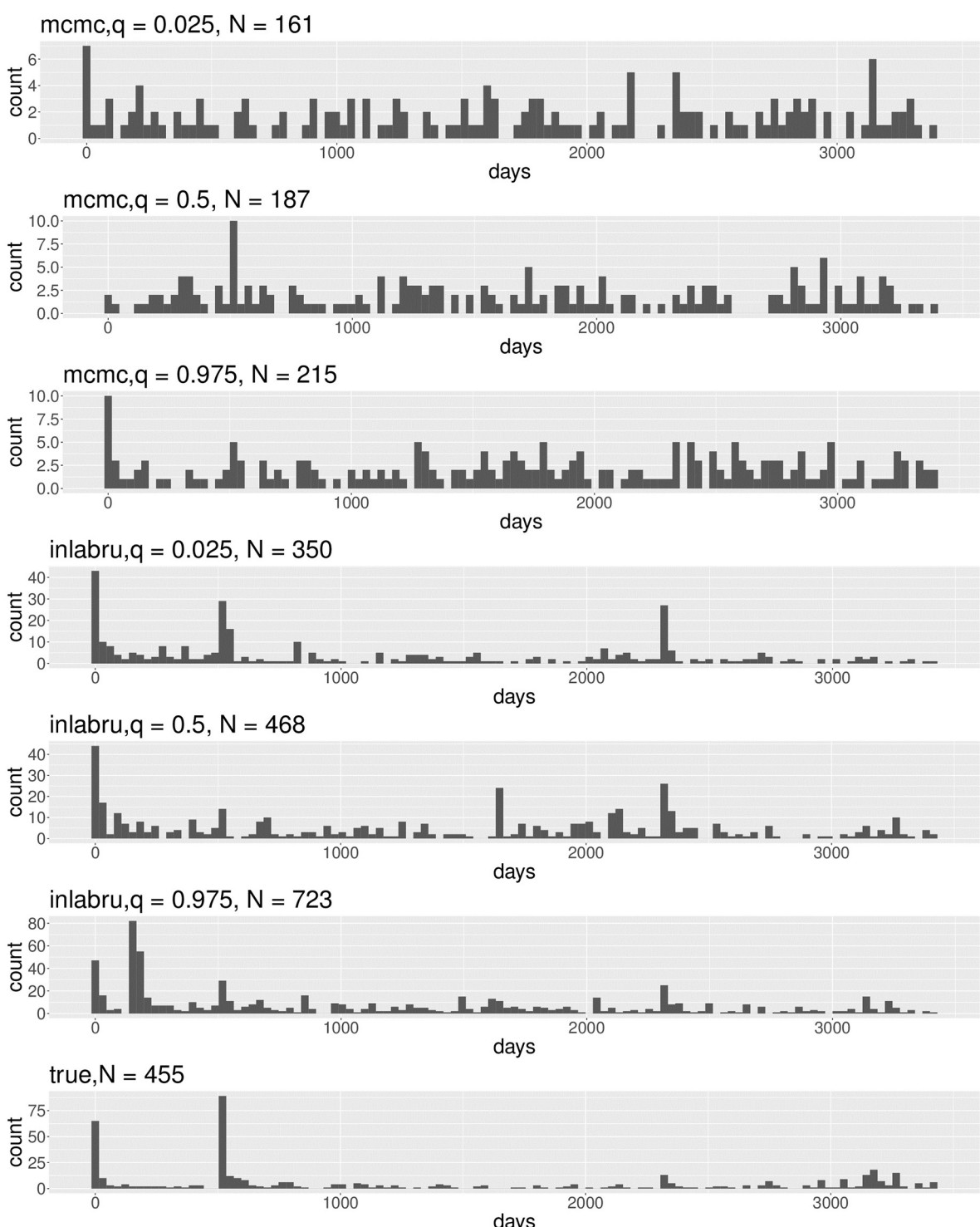

**Fig 6. Histograms of monthly earthquakes in the 0.025, 0.5, and 0.975 quartiles of the distribution of the number of earthquakes in the synthetic earthquake catalogs corresponding to the earthquake catalogs, and the actual earthquake catalogs, respectively, using the results of the INLA and MCMC algorithms fitted to the ETAS model.**

earthquakes in Xinjiang are mainly distributed in Hotan, Kashgar, Kizilsut, Yili, and Boltala, and the research focuses on these 5 regions. Due to the limitations of the number of strong earthquakes in the study area and the computational conditions of the ETAS model, we only estimated the earthquake sequences in the five high seismicity regions in Xinjiang and discussed the temporal characteristics, and the conclusions drawn only reflect some of the statistical characteristics.

The ETAS model is implemented based on the INLAbru package and applied to the earthquake sequences in the above-selected five regions. Comparing the ETAS model parameters of earthquake sequence activity in different regions of Xinjiang, is conducive to the study of the characteristics of each aftershock sequence. The results show that there are obvious regional differences in ETAS model parameters of earthquake sequences in different regions of Xinjiang. Then, we take the Ms7.3 earthquake sequence in Hotan region on February 12, 2014, as an example to investigate the change characteristics of earthquake sequence parameters with time in the short period after the main earthquake. The results show that the model parameters change significantly within 15 days after the main earthquake, in which the mu value of the fit value changes the most, followed by the p-value of the fit value, the K and c values of the fit value change relatively flat, and the alpha value of the fit value shows a sudden increase or decrease within 15 days after the earthquake, after which the model parameters show a relatively stable trend.

To prove the advantages of the INLAbru method and test the goodness of fit of the model, the accumulated earthquake number of the Ms6.0 earthquake series in Hotan region was compared with the fitting curve of the ETAS model achieved by INLAbru and MCMC. Comparing the occurrence time and magnitude size of the earthquake sequences generated by fitting the ETAS model based on the INLAbru and MCMC methods, the occurrence time and magnitude of the earthquake sequences generated by fitting the ETAS model by the INLAbru method are closer to the actual earthquakes. In the case of the same number of earthquake sequences (taking the 2012 Ms earthquake sequence in Hotan region as an example, a total of 455), the calculation time of the statistical MCMC method is 1.48207 minutes, while the INLAbru method only needs 0.29984 minutes to achieve model fitting by relying on the posterior technology of approximate parameters, which greatly reduces the calculation time. The advantages of this method in the face of huge data sets of earthquake occurrence earthquakes will be more obvious.

In the end, the aftershocks 100 days after the Ms7.3 earthquake series in Hotan region are forecasted by using the ETAS model parameter estimation results. The results show that the actual observed earthquakes during the forecasting period are all included in the forecast interval, which can provide a reference for identifying the period of high incidence of earthquake aftershocks after the main earthquake in these regions. At the same time, we used the fitting mean of ETAS model parameters fitted by MCMC and INLA algorithms to calculate the number of long-term forecast earthquakes after the main earthquake and compared them. The results showed that the expected number of earthquakes fitted by the INLA algorithm was closer to the number of observed earthquakes than that fitted by the MCMC algorithm. Compared with the MCMC algorithm, the INLA algorithm can accurately forecast the number of earthquakes in a sudden period, which is more conducive to the forecasting of strong aftershocks.

## Author Contributions

**Conceptualization:** Ke Li.

**Funding acquisition:** Maofa Wang, Huiguo Zhang, Xijian Hu.

**Methodology:** Xijian Hu.

**Software:** Ke Li.

**Writing – original draft:** Ke Li.

**Writing – review & editing:** Ke Li, Huiguo Zhang.

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
