## [Decision Letter · Decision Letter 0]

5 Jan 2024

PONE-D-23-41552Spatio-temporal characterization of seismic sequence parameters and forecasting of strong aftershocks in Xinjiang based on the ETAS modelPLOS ONE

Dear Dr. Hu,

Thank you for submitting your manuscript to PLOS ONE. After careful consideration, we feel that it has merit but does not fully meet PLOS ONE’s publication criteria as it currently stands. Therefore, we invite you to submit a revised version of the manuscript that addresses the points raised during the review process.

We look forward to receiving your revised manuscript.

Kind regards,

Dr. S. M. Anas, Ph.D.(Structural Engg.), M.Tech(Earthquake Engg.)

Academic Editor

PLOS ONE

Journal Requirements:

   "This study was supported by the National Natural Science Foundation of China (11961065, 42164002) , The Ministry of Education of Humanities and Social Science project ( 19YJA910007) , and  the Natural Science Foundation of Xinjiang (2023D01C01)."

3. We note that Figures 1 and 3 in your submission contain map/satellite images which may be copyrighted. All PLOS content is published under the Creative Commons Attribution License (CC BY 4.0), which means that the manuscript, images, and Supporting Information files will be freely available online, and any third party is permitted to access, download, copy, distribute, and use these materials in any way, even commercially, with proper attribution. For these reasons, we cannot publish previously copyrighted maps or satellite images created using proprietary data, such as Google software (Google Maps, Street View, and Earth). For more information, see our copyright guidelines: http://journals.plos.org/plosone/s/licenses-and-copyright.

a. You may seek permission from the original copyright holder of Figures 1 and 3 to publish the content specifically under the CC BY 4.0 license.  

Additional Editor Comments:

Dear Authors,

I am writing to inform you about the outcome of the peer review process for your manuscript titled "Spatio-temporal characterization of seismic sequence parameters and forecasting of strong aftershocks in Xinjiang based on the ETAS model" [PONE-D-23-41552], submitted to PLOS ONE.

The manuscript underwent a thorough review process, and I am pleased to inform you that Reviewer 1 has recommended acceptance. However, Reviewer 2 has provided constructive feedback and recommended major revisions. Reviewer 2 highlighted the need for clarity in interpreting the computational results and emphasized the importance of addressing the various aspects. Furthermore, specific comments and suggestions have been provided to enhance various aspects of the manuscript. After a preliminary analysis of both reviewers' comments and considering the nature of the suggestions, I have decided to proceed with a "Major Revision" decision for your manuscript.

Important Note from the Academic Editor, Dr. S. M. Anas:

I would like to draw your attention to the fact that citing the papers recommended by the reviewers is not obligatory for your revised manuscript. The decision of whether or not to incorporate the suggested papers into your revised version rests entirely with you. While the reviewers have proposed these citations to improve the quality and credibility of your research, the ultimate choice is yours. You are free to opt not to include any of the suggested papers in your revised manuscript if you believe they are not pertinent or do not contribute to the value of your study.

I kindly request you to carefully consider the reviewer’s comments during your revision. Once you have addressed these concerns, please submit the revised manuscript through the journal's online submission system.

Your revised manuscript will undergo a second round of review to ensure it meets the standards of PLOS ONE. Thank you for your understanding and cooperation. If you have any questions or require further clarification, please do not hesitate to contact me.

Best regards,

Dr. S. M. Anas

Academic Editor

PLOS ONE

Reviewers' comments:

Reviewer's Responses to Questions

**Comments to the Author**

1. Is the manuscript technically sound, and do the data support the conclusions?

Reviewer #1: Yes

Reviewer #2: Yes

2. Has the statistical analysis been performed appropriately and rigorously? 

Reviewer #1: Yes

Reviewer #2: Yes

3. Have the authors made all data underlying the findings in their manuscript fully available?

Reviewer #1: Yes

Reviewer #2: Yes

4. Is the manuscript presented in an intelligible fashion and written in standard English?

Reviewer #1: Yes

Reviewer #2: No

5. Review Comments to the Author

Reviewer #1: In this paper, the Integrated Nested Laplace Algorithm (INLA) is applied to the

Epidemic Type Aftershock Sequence (ETAS) model, and the parameters of the ETAS

model are obtained.

Revised version of paper has been improved and it could be accepted as it.

Reviewer #2: The paper deals with the application of ETAS model for revealing spatiotemporal characteristics. Although well elaborated the main message is missing: what is the interpretation of the resulted computations? The physics behind is missing and must be addressed in the revised version of the manuscript.

There are, additionally, certain points in the manuscript that need additional work and corrections. Specific comments are reported, which I hope will contribute to the improvement of its revised version.

GENERAL COMMENTS

1. It would be very convenient for our conversation here, if the lines were numbered

2. In many places throughout the text, some sentences are too long to clearly express and describe either the content or the scope of what is described to be done

3. A careful reading will improve grammar and syntax – it is necessary

4. Please, say earthquakes – why are seismic events, or simply events?

MAJOR COMMENTS

1. Page 12, 1st paragraph: The difference of parameters values among the sequences must be discussed and interpreted appropriately.

2. Page 13, 1st paragraph, 2nd line: “almost the same” are in all similar cases of the published results but there is extensive discussion on the differences as far as the time they occur and their magnitude. Please, discuss and interpret these differences.

SPECIFIC COMMENTS

1. Page 11, 2nd paragraph: “the form of tectonic movements” – do you mean the type of faulting? Or the deformation rates? Please be specific.

2. Page 11, 2nd paragraph: “ … the different types” – same as before. please, clarify what do you mean by “different types of seismic ruptures”

3. Page 11, 2nd paragraph: “ … we can observe … in alpha values” – You need to compare the a – values with the ones obtained in other regions worldwide

4. Page 11, 2nd paragraph: “ … in p values” – Could you interpret the p – values in connection with physical and mechanical characteristics of the activated area? There are plenty relevant publications on that aspect

5. Page 12, 2nd paragraph: “: Could you comment on that and provide interpretation?

6. Figure 6 caption: Say the occurrence year better than the magnitude alone

7. Page 13, 1st paragraph, 1st line: name the figure, give the figure number

8. Page 14, 3rd paragraph: “… credible intervals” – which are these intervals? Hrad to be shown in the graph. You need firstly to name the numbers and then to magnify the part where the values are almost all null.

9. Page 14, 3rd paragraph: “… of the strong aftershock” – when did they occur? How strong? How do you characterize then as “strong”?

10. Page 14, last paragraph: “… selection of the period …” – why July 2, 2023 specifically? You need to explain this choice.

11. Font size on the labels of Figure 7: Hard to be seen – please improve the figure

12. Last paragraph: Please, better erase the: “of the paper”

6. PLOS authors have the option to publish the peer review history of their article (what does this mean?). If published, this will include your full peer review and any attached files.

Reviewer #1: No

Reviewer #2: **Yes: **Eleftheria Papadimitriou

---

## [Author Response · Author response to Decision Letter 0]

14 Feb 2024

Reviewer#2, GENERAL COMMENTS, Concern#1 :

-It would be very convenient for our conversation here, if the lines were numbered

Author response: Thank you very much for your reminder! We have numbered each line in the manuscript, to facilitate better communication with you.

Author action: We have numbered each line in the manuscript, to facilitate better communication with you.

Reviewer#2, GENERAL COMMENTS, Concern#2:

-In many places throughout the text, some sentences are too long to clearly express and describe either the content or the scope of what is described to be done.

Author response: We sincerely thank you for your careful reading. Based on your suggestions, we have made revisions to the parts of the manuscript that have too long sentences and vague descriptions.

Author action: We have made revisions to the manuscript for long sentences and vague descriptions. The location of the revised parts is as follows: Page 1, line 30; Page 2, line 56; Page 2, line 79；Page 2, line 88；Page 3, line 108；Page 4, line 145；Page 6, line 222；Page 7, line 252；Page 9, line 303；Page 10, line 329；Page 13, line 445 (Second,...)；Page 15, line 469. We have highlighted all the modifications.

Reviewer#2, GENERAL COMMENTS, Concern#3 :

- A careful reading will improve grammar and syntax – it is necessary.

Author response: Thank you very much for your advice! We have sought the help of English language professionals and have re-examined the manuscript in detail, revising the sections with grammatical and syntactical problems.

Author action: We have made the following modifications to the parts of the text that have grammatical and syntactical problems. The location of the revised parts is as follows: Page 1, line 16；Page 1, line 18; Page 1, line 26; Page 1, line 37; Page 2, line 50; Page 3, line 101; Page 3, line 131; Page 8, line 297; Page 11, line 365; Page 14, line 461; Page 16, line 494; We have highlighted all the modifications.

Reviewer#2, GENERAL COMMENTS, Concern#4 :

-Please, say earthquakes – why are seismic events, or simply events?

Author response: Thank you for your careful examination. Based on your suggestions, we have revised the words "seismic events" "events" in the text to "earthquakes". 

Author action: We have checked the manuscript and revised the words "seismic events" "events" in the text to "earthquakes". The location of the revised parts is as follows: Page 1, first paragraph, line26；Page 1, first paragraph, line 27；Page 2, line 43, etc. We have highlighted all the modifications.

Reviewer#2, MAJOR COMMENTS, Concern # 1:

-Page 12, 1st paragraph: The difference of parameters values among the sequences must be discussed and interpreted appropriately.

Author response: Thank you very much for your suggestion! Based on your suggestion, we have added a discussion and explanation of the differences between the sequence parameters to the manuscript. The details are as follows: Overall, there were differences in the degree of variation from the sequence parameter values, with the minimum degree of variation in the parameter c and K values, while the degree of variation in the parameter mu, p, and alpha values was very obvious. The reason for the above result is that the parameter c is a very small positive constant, which is mainly considered to include a small constant term in the model equation to avoid a zero denominator. It therefore does not have a very clear physical meaning. It can also be seen from Eq. (2) that the parameters p and alpha are in exponential positions in the model expression, and thus c has less influence compared to p and alpha. Therefore, among the parameters of the ETAS model, the variation of c over time is relatively smooth, while the variation of alpha and p is more obvious.

Author action: We have added a discussion and explanation of the differences between the sequence parameters to the manuscript. The location of the revised parts is as follows: Page 11, line 384.

Reviewer#2, MAJOR COMMENTS, Concern # 2 :

-Page 13, 1st paragraph, 2nd line: “almost the same” are in all similar cases of the published results but there is extensive discussion on the differences as far as the time they occur and their magnitude. Please, discuss and interpret these differences.

Author response: We think this is an excellent suggestion. We have added Fig 4 ("Time-magnitude plots of earthquake sequences generated by the ETAS model and actual earthquake sequences are fitted based on the INLAbru, MCMC method.") to the manuscript, which shows the differences in the occurrence times and magnitude sizes of the earthquake sequences generated by the two methods of fitting the ETAS model, MCMC and INLAbru, and the actual earthquake sequences. Based on Fig 4, we are able to conclude that INLAbru generates earthquake sequences with occurrence times and magnitude sizes closer to actual earthquake sequences compared to the MCMC method.

Author action: We have added a comparison of the occurrence times and magnitude sizes of the earthquake sequences generated by fitting the ETAS model with the actual earthquake sequences by the two methods MCMC and INLAbru to the manuscript. The location of the revised parts is as follows: Page 13, line 438；Page 18, line 541.

Reviewer#2, SPECIFIC COMMENTS, Concern # 1 :

-Page 11, 2nd paragraph: “the form of tectonic movements” – do you mean the type of faulting? Or the deformation rates? Please be specific.

Author response: Thank you very much for your suggestion! We have revised this part as you suggest by changing "the form of tectonic movements" to "different types of faults in each region".

Author action: We have made modifications to this part based on your suggestions. The location of the revised parts is as follows: Page 11, line 339.

Reviewer#2, SPECIFIC COMMENTS, Concern # 2 :

-Page 11, 2nd paragraph: “ … the different types” – same as before. please, clarify what do you mean by “different types of seismic ruptures”

Author response: Thank you very much for your suggestion! We have revised this part as you suggest by changing "… the different types" to "different sequence types of aftershocks".

Author action: We have made modifications to this part based on your suggestions. The location of the revised parts is as follows: Page 10, line 338.

Reviewer#2, SPECIFIC COMMENTS, Concern # 3 :

-Page 11, 2nd paragraph: “ … we can observe … in alpha values” – You need to compare the a – values with the ones obtained in other regions worldwide

Author response: We sincerely appreciate your valuable suggestions! Based on your suggestion, we have added this part to the manuscript to compare our calculated alpha values with the alpha values of the moderate to strong earthquake sequences in mainland China (the distribution of the mean range of alpha is obtained from Ref [37]) .

Author action: We have taken your suggestion to add this part to the manuscript and have added the Ref [37]. The location of the revised parts is as follows: Page 10, line 349.

Reviewer#2, SPECIFIC COMMENTS, Concern # 4 :

-Page 11, 2nd paragraph: “ … in p values” – Could you interpret the p – values in connection with physical and mechanical characteristics of the activated area? There are plenty relevant publications on that aspect

Author response: We are very glad for your suggestion! Based on your suggestion, we have added the physical meaning of the p-value to the manuscript, which characterizes the ability of an earthquake sequence to decay, with a large p-value resulting in a rapid decay of the sequence and a small p-value resulting in a slow decay of the sequence.

Author action: We have taken your suggestion to add this part to the manuscript. The location of the revised parts is as follows: Page 10, line 356.

Reviewer#2, SPECIFIC COMMENTS, Concern # 5 :

-Page 12, 2nd paragraph: “: Could you comment on that and provide interpretation?

Author response: Thank you very much for your suggestion, we have added comments and explanations to this part. Comments and explanations added to the manuscript are listed below: The reason for the "sudden" changes in the sequence parameters can be analyzed from the earthquake sequence activity, which may be due to the occurrence of aftershocks of larger magnitude at the early stage after the main earthquake, thus the earthquake sequence has a large change in the decay rates of aftershock and the excitation degree of aftershock. However, this kind of sequence activity is only a representation analysis, and the adjustment of the stress field in the physical source area or the adjustment and change of the aftershock rupture mechanism needs to be further studied.

Author action: We have added comments and explanations to this part. The location of the revised parts is as follows: Page 12, line 400.

Reviewer#2, SPECIFIC COMMENTS, Concern # 6 :

- Figure 6 caption: Say the occurrence year better than the magnitude alone

Author response: Thank you very much for your suggestion, we have revised the title of Fig 5 to show the year of occurrence of the earthquake sequence. Specifically, we have revised the title of Fig 5 to "Retrospective forecasting test (using the earthquake sequence from February 12, 2014, in Hotan region as an example)".

Author action: We have revised the title of Fig 5 of the manuscript. The location of the revised parts is as follows: Page 15, figure 5, first line of title.

Reviewer#2, SPECIFIC COMMENTS, Concern # 7 :

-Page 13, 1st paragraph, 1st line: name the figure, give the figure number

Author response: Thank you very much for your suggestion! Based on your suggestion, we have given the number of this part of the figure as well as the title.

Author action: We have given the number of this part of the figure as well as the title. The location of the revised parts is as follows: Page 12, last paragraph, first line.

Reviewer#2, SPECIFIC COMMENTS, Concern # 8 :

-Page 14, 3rd paragraph: “… credible intervals” – which are these intervals? Hard to be shown in the graph. You need firstly to name the numbers and then to magnify the part where the values are almost all null.

Author response: Thank you very much for your suggestion! Based on your suggestion, we have described "credible intervals" in the title of Fig 5 with the following title: Fig 5 (top), the black dots indicate the number of earthquakes observed in each forecast period, the red solid line indicates the median number of earthquakes in the synthetic catalog for each forecast period, and the orange areas indicate the 95% forecast intervals for the number of earthquakes in each time. The extreme values for each interval are the 2.5% and 97.5% quantiles of the number of earthquakes in the synthetic catalog that make up the daily forecast. At the same time, we have added Fig 5 (below) giving the logarithm of the number of earthquakes, thus amplifying the almost blank part of this value.

Author action: We have shown "credible intervals" in the title of Fig 5 (top). At the same time, we have added Figure 5 (below). The location of the revised parts is as follows: Page 15, figure 5, title, fifth line.

Reviewer#2, SPECIFIC COMMENTS, Concern # 9 :

-Page 14, 3rd paragraph: “… of the strong aftershock” – when did they occur? How strong? How do you characterize then as “strong”?

Author response: Thank you very much for your suggestion! Based on your suggestion, we have added a detailed description of the "… of the strong aftershock” , here is Ms7.3 ( Hotan, February 12, 2014).

Author action: We have added a detailed description of the "… of the strong aftershock.” The location of the revised parts is as follows: Page 15, line 482.

Reviewer#2, SPECIFIC COMMENTS, Concern # 10 :

-Page 14, last paragraph: “… selection of the period …” – why July 2, 2023 specifically? You need to explain this choice.

Author response: Thank you very much for your suggestion! Based on your suggestion, we have added a reason for the "… selection of the period …". We have chosen the earthquake sequences from this region and time because of the large magnitude of the earthquakes and the long duration of the sequences, which therefore contain a large number of more complete earthquakes.

Author action: We have added a reason for the "… selection of the period …". The location of the revised parts is as follows: Page 16, line 489.

Reviewer#2, SPECIFIC COMMENTS, Concern # 11:

-Font size on the labels of Figure 7: Hard to be seen – please improve the figure

Author response: Thank you very much for your suggestion! Based on your suggestions, we have made revisions to Fig 6 in our manuscript, specifically revising the font size in the figure to become clearer.

Author action: We have made revisions to Fig 6 in our manuscript. The location of the revised parts is as follows: Page 17, figure 6.

Reviewer#2, SPECIFIC COMMENTS, Concern # 12 :

-Last paragraph: Please, better erase the: “of the paper”

Author response: We sincerely appreciate your careful reading. Based on your suggestions, we have deleted "of the paper" in the last paragraph.

Author action: We have deleted "of the paper" in the last paragraph. The location of the revised parts is as follows: Page 18, line 552.

---

## [Decision Letter · Decision Letter 1]

27 Mar 2024

Spatio-temporal characterization of earthquake sequence parameters and forecasting of strong aftershocks in Xinjiang based on the ETAS model

PONE-D-23-41552R1

Dear Dr. Hu,

We’re pleased to inform you that your manuscript has been judged scientifically suitable for publication and will be formally accepted for publication once it meets all outstanding technical requirements.

Kind regards,

Dr. S. M. Anas, Ph.D.(Structural Engg.), M.Tech(Earthquake Engg.)

Academic Editor

PLOS ONE

Additional Editor Comments (optional):

Dear Corresponding Author and Co-Authors,

I hope this email finds you well. I am writing to inform you of the decision regarding the revised manuscript entitled "Spatio-temporal characterization of earthquake sequence parameters and forecasting of strong aftershocks in Xinjiang based on the ETAS model" [PONE-D-23-41552R1], which you submitted to PLOS ONE.

Initially, the revised manuscript was assigned to a previous reviewer who unfortunately declined the review invitation. Consequently, I invited a third reviewer to assess both your responses to the previous reviewers' comments and the revised manuscript. I am pleased to inform you that the third reviewer has expressed satisfaction with your responses and recommended the revised version for publication.

Upon reviewing the reviewers' recommendations and conducting a preliminary assessment of the manuscript myself, I have decided to accept the manuscript for publication in PLOS ONE, subject to the approval of the editorial board.

Please note that this decision is pending final approval from the editorial board. Once approved, you will receive further instructions regarding the publication process.

Once again, congratulations on this achievement, and thank you for choosing PLOS ONE as the venue for disseminating your research findings.

Should you have any questions or require further assistance, please do not hesitate to contact me.

Best regards,

Dr. S. M. Anas

Academic Editor

PLOS ONE

Reviewers' comments:

Reviewer's Responses to Questions

**Comments to the Author**

1. If the authors have adequately addressed your comments raised in a previous round of review and you feel that this manuscript is now acceptable for publication, you may indicate that here to bypass the “Comments to the Author” section, enter your conflict of interest statement in the “Confidential to Editor” section, and submit your "Accept" recommendation.

Reviewer #3: All comments have been addressed

2. Is the manuscript technically sound, and do the data support the conclusions?

Reviewer #3: Yes

3. Has the statistical analysis been performed appropriately and rigorously? 

Reviewer #3: Yes

4. Have the authors made all data underlying the findings in their manuscript fully available?

Reviewer #3: Yes

5. Is the manuscript presented in an intelligible fashion and written in standard English?

Reviewer #3: Yes

6. Review Comments to the Author

Reviewer #3: (No Response)

7. PLOS authors have the option to publish the peer review history of their article (what does this mean?). If published, this will include your full peer review and any attached files.

Reviewer #3: No

---

## [Editor Report · Acceptance letter]

3 Apr 2024

PONE-D-23-41552R1 

PLOS ONE

Dear Dr. Hu, 

I'm pleased to inform you that your manuscript has been deemed suitable for publication in PLOS ONE. Congratulations! Your manuscript is now being handed over to our production team.

Kind regards, 

on behalf of

Dr. S. M. Anas 

Academic Editor

PLOS ONE